# Technology and Properties of Peripheral Laser-Welded Micro-Joints

**DOI:** 10.3390/ma14123213

**Published:** 2021-06-10

**Authors:** Szymon Tofil, Hubert Danielewski, Grzegorz Witkowski, Krystian Mulczyk, Bogdan Antoszewski

**Affiliations:** Laser Research Centre, Faculty of Mechatronics and Mechanical Engineering, Kielce University of Technology, Al. Tysiąclecia Państwa Polskiego 7, 25-314 Kielce, Poland; hdanielewski@tu.kielce.pl (H.D.); gwitkowski@tu.kielce.pl (G.W.); kmulczyk@tu.kielce.pl (K.M.); ktrba@tu.kielce.pl (B.A.)

**Keywords:** laser micro-welding, cryosurgical probe, butt weld, thin tube welding

## Abstract

This article presents the results of research on the technology and peripheral properties of laser-welded micro-couplings. The aim of this research was to determine the characteristics of properly made joints and to indicate the range of optimal parameters of the welding process. Thin-walled AISI 316L steel pipes with diameters of 1.5 and 2 mm used in medical equipment were tested. The micro-welding process was carried out on a SISMA LM-D210 Nd:YAG laser. The research methods used were macroscopic and microscopic analyses of the samples, and assessment of the distribution of elements in the weld, the distribution of microhardness and the tear strength of the joint. As a result of the tests, the following welding parameters are recommended: a pulse energy of 2.05 J, pulse duration of 4 ms and frequency of 2 Hz, beam focusing to a diameter of 0.4 mm and a rotation speed of 0.157 rad/s. In addition, the tests show good joint properties with a strength of more than 75% of the thinner pipe, uniform distribution of alloying elements and a complex dendritic structure characteristic of pulse welding.

## 1. Introduction

The progressive development of technology in many fields is associated with miniaturisation. This is especially true in areas such as electrical engineering, automation, the automotive industry, aerospace and medical technology. In many solutions, there is a need for durable and reliable joints. An effective way to meet these expectations is laser micro-welding. The term is not unambiguous, but it is usually assumed that the term refers to a welding process in which the welded parts and the penetration zone have a dimension not exceeding 1 mm [1]. The literature on micro-welding addresses such topics as the following:− Micro-welding of materials with diverse properties [2,3,4,5,6,7];− Choice of laser type and the impact of the operating parameters on the joint properties [1,7,8,9];− Micro-welding of materials with diverse shapes and sizes [2,3,4];− Modelling of the micro-welding process [1,10,11,12,13,14];− Evaluation of properties and metallurgical changes of laser-welded micro-joints [15,16,17].

Micro-scale laser welding has specific features that make the process require a special approach. Here, we are dealing with a material with a very low heat capacity, and therefore the energy supplied must be extremely precisely selected. Laser welding technology, as opposed to standard methods, is distinguished by features that make laser welding, on a micro-scale, the best possible solution next to electron-beam welding [3,4,9]. This is determined by the possibility of choosing the type of laser beam used for welding, the possibility of precise energy dosing in the form of pulses of a short duration and the possibility of focusing the beam on a very small area located in a specific place. Nd:YAG, fibre and disc lasers are the most commonly used lasers for micro-welding applications [1,18,19,20,21]. The current research interest is focused on the use of ultrashort pulse lasers [6,9]. The nanosecond and picosecond interaction of pulses causes a qualitative change in the phenomena occurring in the material and is a rewarding research area, which also enters the field of micro-welding. The above-mentioned considerations determine that each case of micro-welding, depending on the requirements set, should be treated with special care for the precise selection of all aspects of the technological process.

Macro-scale welding of stainless steels is widely reported in the literature. However, welding on a micro-scale causes significant differences. The progress of miniaturisation contributes to extensive progress in micro-welding, including laser welding of stainless steels, which is reflected in the number of articles on this subject. The authors of [22] studied the welding parameters for butt welding of 100 μm-thick foil of AISI 316 steel. Good quality joints were obtained using a pulsed Nd:YAG laser for the following process parameters: energy of 1.75 J, welding speed of 525 mm/min and pulse duration of 4 ms. According to the results of the studies in [23,24,25,26,27,28,29,30,31], the increase in laser power increases the depth of the weld when welding stainless steel sheets. In [31], an effective empirical model was presented in order to predict the joint properties of stainless AISI 304 and 316 steels, where an estimation error between the model and experiment equal to 15% was achieved. The laser power and scanning speed were indicated as essential factors affecting the joint strength. The grain refinement together with the formation of δ-ferrite in the weld causes a slight increase in hardness in the fusion zone. Similar results were observed in [32,33,34]. The authors of [7] carried out extensive research on butt welding of thin sheets (0.45 mm) of AISI 316 steel with a Nd:YAG laser. The study confirmed that the laser power is the most important parameter in the welding of thin sheets because it significantly affects the properties of mechanical and metallurgical joints. The overlap of pulses significantly affects the surface roughness and strength of welded joints. The authors of [32] presented the results of a comparative study of arc and Nd:YAG laser welding of AISI 316 steel. It was found that laser welding causes less distortion and lower residual stress compared to arc welding. The differences between laser macro- and micro-welding of stainless steels refer mainly to the narrower range of micro-laser welding parameters, which guarantees a good joint quality [22,25,29]. In addition, with micro-welding, welds have a very narrow heat-affected zone (HAZ) along the fusion line, and microstructure changes are related to grain refinement during crystallisation [10,13,30]. Therefore, hardness changes are practically negligible. A significant barrier to achieving an acceptable level of welded joints is thermal deformation, which can cause crack formation during weld formation and lead to loss of its continuity [27,28,29,30]. In the study of [35], the authors studied, in detail, the problem of gaps in lap joint welding, where, by studying the changes in the microstructure and the shape of the weld, they recommended an acceptable gap size of less than 0.1 mm. One of the main problems presented in the abovementioned publications is the problem of stress concentration and deformation of welded materials, which is particularly troublesome in circumferential lap welding [35]. As a result of deformation, the gap between the welded elements may increase, and defects may appear in the joint. For these reasons, the problem of joined element stabilisation often determines the effect of the entire operation.

While the welding of thick-walled pipes is widely described in the literature, there are no reports on the welding of thin-walled AISI 316 steel pipes with diameters on the order of millimetres.

Of special interest in recent years are medical subjects [2,6] in which laser technology is applied, such as stents, implants and dentures, probes and other medical components.

The authors of this article deal with the development of the technology of laser micro-welding of elements of a medical implement, in a design solution with basically three types of circumferential welds (Figure 1—own photos produced by the authors of this article)—lap–butt weld of a probe tip with a tube-shaped probe body; lap joint of two tubes; and fillet joint of two tubes.

The main purpose of this research team is to develop a laser welding method for thin-walled tubes which can be used as an alternative to brazing when the following requirements need to be achieved. The joints shall have a weld with no visible welding imperfections, and a regular face and back of the weld with no traces of surface scorching. Small single pores are acceptable. The weld should not degrade the strength of the probe (75% of the strength of the weaker probe component) or its corrosion resistance. The face of the weld should not extend above the parts being joined.

The developed laser micro-welding technology is innovative for the target product and can replace solutions in current use. For the implementation of the established task, it was necessary to solve the following problems: the selection of a laser device and its operating parameters, the study of mechanical properties of the joint and the assessment of metallurgical changes in the joint area.

## 2. Experimental Tests

### 2.1. Methodology

Pulse welding was adopted as the welding method due to the small size and heat capacity of the welded parts. This provided protection to the component from excessive temperature increases that could cause structural changes and oxide formation on the surface. The micro-welding process was performed using a typical process parameter selection procedure with a professional SISMA LM-D210 laser micro-welder. In accordance with PN-EN ISO 13919-1, the correctness of the joint structure was determined, and welding imperfections were identified. Welding was performed in two variants with an argon shroud and without a gas shield. The chemical composition of the welded material is shown in Table 1. The presented paper describes a detailed study on the selection of welding process parameters and joint properties for a lap weld according to Figure 1B. The test procedure for the other welds followed the same pattern, and due to the limited volume of this article, the results for the other two types of welds are not included here.

The completed joints were continuously evaluated by visual tests (VT) and qualified for further testing. The planned tests included: macroscopic and microscopic tests of the weld (HIROX KH-8700 microscope), microstructure and elemental distribution with an accelerated voltage of 15 keV and a work distance of 10 mm according to ISO 22309:2011 (JOEL 7100f electron microscope and HIROX KH-8700 microscope), and mechanical property tests with a standard repeatable measurement accuracy of less than 2.5 N (INSTRON 4502 device in accordance with the recommendations of PN-EN ISO 6892-1). The strength of the resulting welded joints was determined by static tensile testing. The repeatability of the results was checked by producing two sets of samples for welding in an argon shroud and without a shielding gas. Five samples were taken for each set based on the process parameters for set 4 as shown in Table 2. The test samples were mounted in specially designed mounting fixtures to maintain the alignment of the tensile force. The strength tests were conducted at a constant tensile speed of 0.5 mm/s for the samples.

Microhardness tests were conducted with load HV0,1 (Innovatest Nexus 4303 hardness tester in accordance with PN-EN ISO 6507-1). Measurements were recorded for samples welded in an argon shroud in the tube core material, in the centre of the penetration and at its edge.

Specimens that qualified for microscopic tests were embedded in resin and then finely ground to visualise the penetrations. The penetrations were first observed on a HIROX microscope, where the geometry of the characteristic zones was measured, as shown in the diagram (Figure 2). The width of the face (A), the depth of the penetration zone (B) and the width of the penetration at the boundary of the joined parts (C) were determined here.

### 2.2. Technological Experiment

Drawn tubes made of AISI 316L-grade steel with nominal diameters of 1.5 × 0.15 and 2 × 0.2 were used for the experiment. The chemical composition of the welded material is shown in Table 1.

The dimensions of the workpieces to be welded allow the smaller-diameter tube to be inserted into the larger-diameter tube concentrically with a nominal clearance of 0.05 mm. Allowing for alignment errors and tube circularity, this clearance may vary locally in the range of 0 to 0.1 mm. Actual measurements of the tube series showed that the mean value of the inner diameter of the thicker tube was 1.618 mm, with a standard deviation of 0.010 mm and a spread of 0.024 mm, and the mean value of the outer diameter of the thinner tube was 1.608 mm, with a standard deviation of 0.008 mm and a spread of 0.029 mm. Such an arrangement makes it possible to perform a circumferential lap weld without having to support a second tube after the first tube is placed in the turnover fixture. At the same time, it is consistent with the recommendations available in the literature [21] according to which the allowable gap at lap joints should not exceed 25% of the spot diameter. The schematic diagram of the prepared joint is shown in Figure 2.

A test bench equipped with a Sisma LM-D210 laser micro-welder illustrated in Figure 3 was used to conduct the experiment. The device is based on a Nd:YAG pulsed laser emitting radiation of 1064 nm wavelength. The emitted beam was multi-mode. The laser was equipped with an integrated focusing and observation system with a camera and monitor. This arrangement enables a focal spot with an adjustable diameter of 0.1–2 mm. The device has digital adjustment of laser operating parameters, such as pulse duration, energy per pulse and pulse generation frequency. Pulse duration can be varied from 0.1 to 25 ms, pulse energy from 0 to 210 J and frequency from 0 to 50 Hz. The maximum power per pulse reaches 10.5 kW. A protective atmosphere in the weld area was provided by local argon flow.

Rotation of the workpiece around the axis during the welding process was performed using a miniature speed-controlled turnover fixture developed and made for this purpose. Rotational motion was provided by a stepping motor with microstep control. The motor controller was controlled by a PC and allows a speed range of 0.002–20 rad/s.

In order to select the optimum process parameters, preliminary experiments were conducted by performing a series of circumferential welds using different laser beam pulse energies in the range of 1.36–2.27 J. Based on the preliminary tests, the pulse duration and frequency were selected to be 4 ms and 2 Hz, respectively. The criterion adopted for the selection of the parameters was to achieve the effect of applying individual pulses at the level of 80% while maintaining the correct appearance of the weld face and the absence of clear thermal interaction in the weld area. The recommended overlap value for tight joints in the literature [21] is 80 to 90%. The focus diameter of the laser beam used in the welding process was 0.4 mm. The workpiece speed was 0.157 rad/s. The beam was quenched after the welded workpiece had rotated through an angle of 6.3 rad to ensure a continuous circumferential weld. Table 2 lists the welding parameters for which a positive evaluation was obtained from visual testing.

## 3. Properties of Laser Micro-Joints—Results and Discussion

### 3.1. Microstructural Tests

After producing metallographic sections, the images of penetrations were revealed, which are shown in Figure 4. It can be observed here that the penetration depth (Figure 2B) of welds 1, 2 and 3 (Figure 4) is less than half the thickness of the inner tube. Moreover, in the case of weld 1, there was no fusion penetration into the lower material (inner tube); therefore, we deal with the so-called adherence. At the same time, there is a clear increase in the depth of penetration with increasing pulse energy. In the case of weld 5, there is a penetration of both materials to be welded, but there was an outflow of material from the root side and a significant concavity of the weld face. Based on observations of the structure of the welds obtained, weld 4 was selected, which is characterised by a lack of buckles of the weld face, fusion penetration into the lower tube for at least half of its thickness and the absence of visible welding imperfections.

The appearance of the face as well as its width (Figure 2A) for all welds (except weld 5) is to be qualified as correct. The occurrence of a slight face concavity of welds 3 and 4 is acceptable in contrast to the occurrence of a convex face. The fusion penetration width (Figure 2C) at the boundary of the joined tubes increases with increasing pulse energy. It can be seen that the value of width A stabilises for welds 3, 4 and 5. The face width for weld 4 was 0.468 mm. The trends of changes in the A, B and C quantities characterising the penetration geometry are shown in the graph in Figure 5.

The conducted tests showed that the weld width in the overlap zone (C) has the highest growth dynamics. Additionally, the weld depth (B) increases with increasing pulse energy, which is natural for the process tested. On the other hand, the weld face width (A) increases slightly due to the constant spot diameter during the performed process. The increase in the weld face width is caused by an increase in the amount of energy transferred to the surface of the material and thus the heating of the welded parts.

A joint quality level analysis was performed on the sample joints produced by pulsed laser micro-welding. Samples were taken from the circumferential lap joint and used for metallographic tests after grinding and etching processes (Figure 4).

Based on the results obtained, it was found that there is a lack of regularity in the structure of welds 1 and 5 (areas marked in red), and in the case of the first weld, the fusion penetration line is located at the boundary of the zone between the welded materials, which indicates an unstable joint in the form of incomplete fusion.

In the case of the fifth weld, we are dealing with a through penetration, but the amount of energy supplied by the laser beam was too high and a concavity of the weld face occurred due to an excessive outflow of material in the root area. Both of the abovementioned welding imperfections disqualify the resulting joints. For joints 2–4, the assumed fusion penetration into the lower material was achieved; however, due to the obtained fusion penetration depths, joint 4 was chosen for further analysis, where a fusion penetration of the lower material above half of its thickness was obtained (Figure 6).

The obtained weld has a correct structure with a slight concavity of the weld face (area 1), which is 20 µm, but it does not exceed the permissible 5% of the thickness of welded materials. Area 2 shows the fusion penetration line of the weld into the base material (Figure 7).

Within area 3, the presence of a heat-affected zone was identified, and no clear zone was found, which is typical of welds in austenitic stainless steels. In the case under consideration, no growth of austenite grains was observed, only elongated ferrite grains forming a discontinuous grid around the austenite grains. Such areas occur in steels with a base material structure consisting of austenite with ferrite δ. At high temperatures, a γ → δ transformation occurs along the fusion penetration line, which begins in the existing ferrite δ grains and progresses toward the area of increased chromium concentration. Upon recooling, this area does not reach phase equilibrium, by which the proportion of ferrite δ increases, forming a narrow heat-affected zone (Figure 7). The weld root is also marked within area 3 along with the overlapping of the individual weld layers resulting from the pulsed welding mode. In pulsed welding, a series of overlapping pulses causes the material to instantaneously melt and recrystallise, producing the characteristic scaly weld bead. A dendritic structure is formed in the interaction area of the individual pulses, but for individual heat cycles, when the laser beam penetrates the lower sheet region only to a certain depth, the impact of the flow field is evident in the solidified structure and the direction of dendrite growth. In the case of partial weld penetration at the pulsed mode of the heat source, discrete growth bands are present, suggesting strong fluctuations in the flow field and affecting the grain growth process and direction.

For the weld case considered, no significant welding imperfections were found; the slight concavity of the weld face is within the acceptable range, and the penetration depth of the laser beam for joint 4 exceeds half the thickness of the lower material. No other type of welding imperfections were found; thus, the fabricated joint was classified as quality level B. Analysis of the microstructure showed a dendritic structure with visible banding of the dendrite growth resulting from the pulsed welding mode. A narrow heat-affected zone characteristic of austenitic steel was identified.

For stainless steels, especially for surgical applications, corrosion resistance is an important aspect, which is ensured by an appropriate content of alloying elements in the form of chromium and nickel as well as by the reduction in carbon in the alloy. Additionally, in the case of the considered material—austenitic stainless steel of 316L grade—molybdenum is introduced into the alloy composition, which improves the steel’s resistance to corrosion, especially intercrystalline corrosion. During welding, the content of alloying elements in the weld may change, which may adversely affect the properties of the joint, including corrosion resistance. A preliminary evaluation of the joint in this regard can be conducted by performing X-ray energy dispersion spectrum analyses for the indicated elements. Analysis of the element distribution was performed for area 2 by means of which the quantitative content of selected elements was determined in defined measuring points for the joint made without a protective atmosphere (Figure 8A) and in an argon shroud (Figure 8B).

According to the measurement points marked above, a quantitative analysis of the chemical composition of selected alloying elements was performed for the exemplary spots, which is shown in Table 3.

The quality level of a joint in terms of anti-corrosivity is defined not only by the content of selected elements within the weld but also by the difference in the content of these elements in relation to the base material. In order to determine if there are significant differences between the distribution of chromium, nickel, molybdenum and iron in the weld relative to the base material, qualitative analysis was performed on the cross-section of the joint (Figure 9).

The results of the analysis show a uniform distribution of the selected elements, close to linear in nature, and no significant differences between the nickel and molybdenum contents in the weld relative to the base material for both welding variants. On the other hand, a slight reduction in chromium was observed, which may be due to the migration of this element during metallurgical processes in connection with the residual content of atmospheric oxygen in the zone between welded elements, or to the burning out of a part of chromium during the welding process. A similar trend was observed for both cases. However, no separations were found in the penetration zone, especially in the chromium carbides, which may cause intercrystalline corrosion, and it was therefore assumed that the welded joint is characterised by good anti-corrosion properties. A full evaluation of the corrosion resistance of the joint will be obtained after detailed specialised corrosion testing.

### 3.2. Microhardness and Joint Strength Testing

Hardness measurements were performed according to the scheme shown in Figure 10. The measured microhardness values for welds (Figure 4) 3 and 4 averaged 371 HV0.1 in the tube core material (1), ranging from 220 HV0.1 in the central part of the penetration (2) to a value of 240 HV0.1 near the fusion penetration line (3). This variability is due to the different cooling conditions of the molten material. The increased microhardness value of the tube core is the result of crumple occurring during the tube drag process. The result obtained shows that there are no overheating effects on the material, which is also confirmed by the absence of discolouration of the material after the welding process. For samples without the argon shroud, there are no significant differences in the microhardness measurements recorded.

### 3.3. Test Results for Joint Strength

The results of tensile shear test are shown in Table 4 and in Figure 11.

The results obtained were statistically analysed to determine the mean, minimum and maximum values and the mean dispersion of results. On the basis of the analysis performed, the mean value of the breaking force of the welded joints made was determined, which is 754 N for the weld made without the application of a shielding gas and 770.6 N for the joints made in an argon shroud. A slight increase of only 2.20% in the average strength of joints made in an argon shroud was observed. However, taking into account the dispersion of the results, it cannot be clearly stated that the shielding gas contributes to the increase in the joint strength. In view of the fact that the measured breaking strength of the thinner tube in the joint is 976 N, it must be concluded that the required strength criterion of the joint (75% of the strength of the weaker element in the joint) was met. This condition is met for both unshielded and argon-shielded welding.

During the static tensile test, a rupture characteristic of brittle fracture occurred in each of the welds tested. Until the moment of breaking, the samples underwent a characteristic deformation passing successively through the areas of elasticity and plasticity. No obvious plasticity growth before fracture was observed. The course of deformation in both highlighted areas should be described as typical for austenitic steel. Tensile cracking of a sample is the result of the joint interaction of tensile and shear stresses. There are two types of tensile cracking in laser welds: heat-affected zone boundary cracking and interfacial cracking. Since the hardness of the tube material is greater than the hardness of the weld, the tensile deformation is concentrated in the heat-affected zone or at its boundary. The grains in the heat-affected zone tend to grow because the heat-affected zone is kept at a high temperature for a long time; the high temperature gradient results in the transformation of the material structure. Hence, the heat-affected zone is often the weakest area of the weld. As a result, joint breaking most often occurs along the fusion penetration line or at the boundary between the heat-affected zone and the base material. The fracture characteristics observed (for sample nr 4) in the scanning microscope images are shown in Figure 12. These observations suggest that the fractures are complex in nature. On the one hand, it was noticed that fractures along the fusion penetration line were observed on some samples (Figure 12A). On the other hand, some samples, in whole or in part, were fractured across the penetrated zone (Figure 12B). In both observed mechanisms (Figure 12A,B), areas of brittle fracture (marked 1) and areas of ductile fracture (marked 2) are distinguished. At the fusion penetration line crack, traces of overlapping consecutive laser pulses are visible, on which transverse cracks are visible (Figure 12C). They are the result of accumulated stresses resulting from tensile and shear forces as well as internal stresses in the exposed fusion penetration line. Figure 12D shows the ductile fracture on the inner edge of the welded tubes. A partially developed ductile fracture is observed in the inner part of the tube outside the penetration (Figure 12E). Figure 12F shows a fracture along the fusion penetration line in the beam extinction area at the end of the circumferential weld. The authors believe that the complex nature of the fracture is the cause of the large dispersion of shear fracture strength results. This is due to the increased gap between the joined elements. It results from deformations during welding and the accuracy of welded tubes. This proves that the weld may not be uniform along the entire circumference. To avoid this, a dimensional selection of the tubes should be made before welding.

The shape shown in microphotograph 12f is the result of the last laser pulse and is a typical image for extinction without reducing the laser power (without power ramping). The solidification of the metal here takes place from the area of the highest temperature gradient, i.e., from the outer diameter, and the material in the liquid phase begins to crystallise at a speed higher than the flow velocity of the liquid metal; hence, the centre is only partially flooded, which results in the formation of a concave.

## 4. Summary

The results of the conducted tests on laser overlap welding of thin-walled tubes with nominal diameters of 1.5 × 0.15 and 2 × 0.2 mm allow for the formulation of the following conclusions.

1. The best weld properties were obtained in argon-shielded welding at a pulse energy of 2.05 J, pulse duration of 4 ms and frequency of 2 Hz, beam focusing to a diameter of 0.4 mm and a rotation speed of 0.157 rad/s.

2. The weld face is smooth with a gentle undulation corresponding to the number of pulses on the circumference, avoiding the formation of surface oxides when welding in an argon shroud.

3. The fusion penetration zone has a typical U shape, and its microstructure is characterised by the occurrence of dendrite density in the zone near the fusion penetration boundary. Moreover, the boundaries of the overlapping consecutive coagulated melting fronts of the material stand out in the fusion penetration structure.

4. The cracking of samples during static tensile testing is complex in nature. It most often occurs along the fusion penetration line, and less often by penetration. The observed fractures show areas of brittle fracture and weaker-shaped areas of ductile fracture.

5. Analysis of the chemical composition according to the defined measurement points showed no significant differences for both welding variants. No excessive burning or oxidation of alloying elements was observed in both analysed welds relative to the base material.

6. The linear nature of the chromium, nickel and molybdenum distribution between the base material and the weld shows the high level of purity of the welds obtained and the lack of reduction in the content of alloying elements. Additionally, no changes were found in the fusion penetration line area, and thus the corrosion protection capability of the material was maintained.

7. Before welding, it is recommended that a dimensional selection of the tubes should be conducted ensuring the smallest possible gap between the welded elements (maximum 0.1 mm).

## Figures and Tables

**Figure 1 materials-14-03213-f001:**
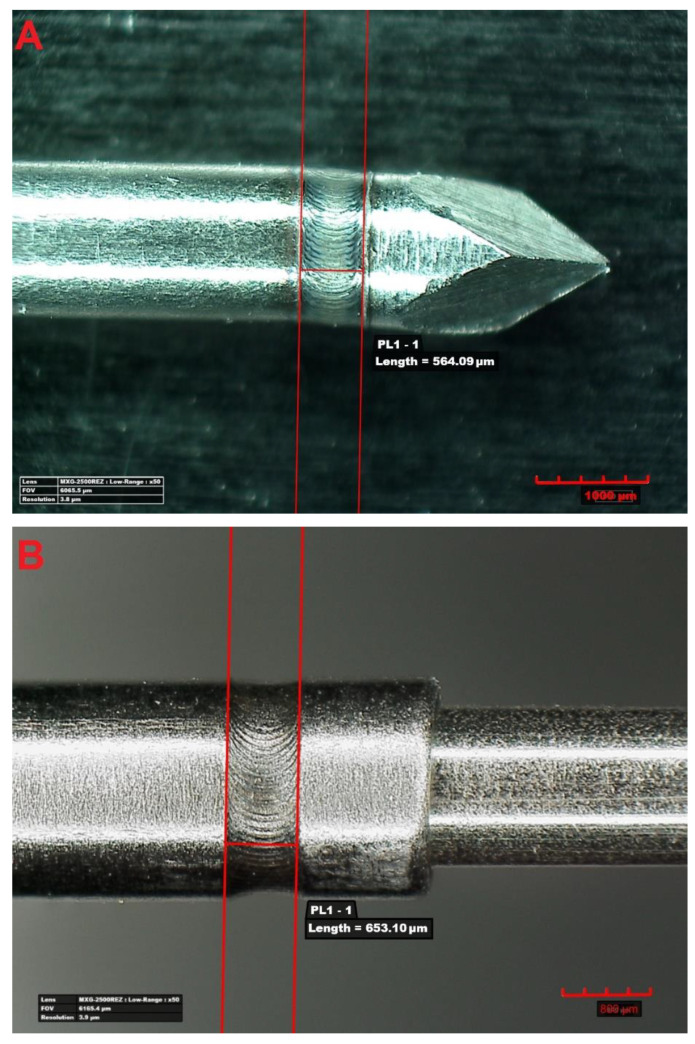
Example view of analysed welds: (**A**) lap–butt weld of the probe tip with a tube-shaped probe body; (**B**) lap joint of two tubes; (**C**) fillet joint of two tubes.

**Figure 2 materials-14-03213-f002:**
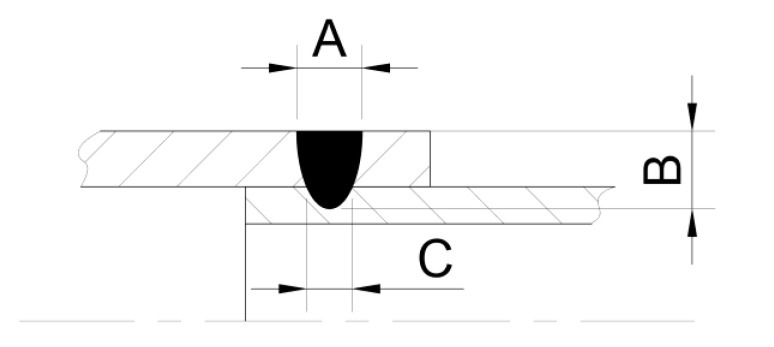
Schematic diagram of the analysed weld.

**Figure 3 materials-14-03213-f003:**
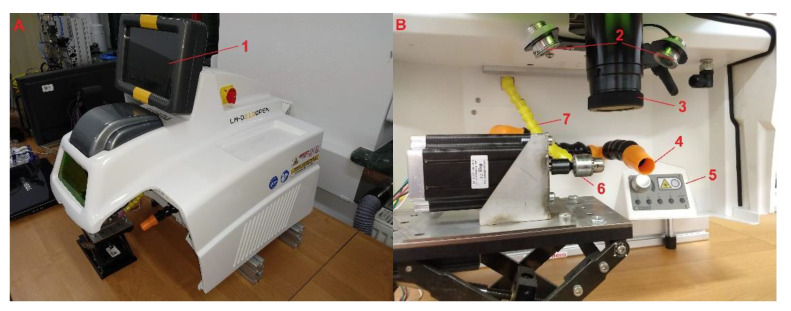
View of the SISMA LM-D210 laser bench. (**A**)—general view, (**B**)—workspace view. 1—front touch panel, 2—lighting spots, 3—laser head, 4—gas extractor, 5—manual parameter set panel, 6—rotary motor, 7—nozzle of gas shield.

**Figure 4 materials-14-03213-f004:**
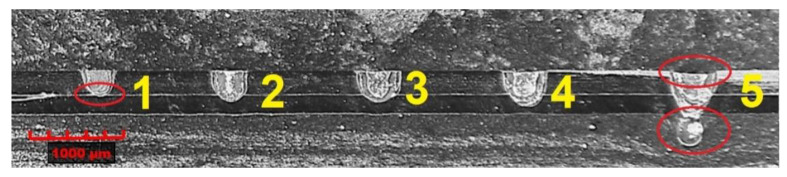
View of obtained welds for lap joints in cross-section.

**Figure 5 materials-14-03213-f005:**
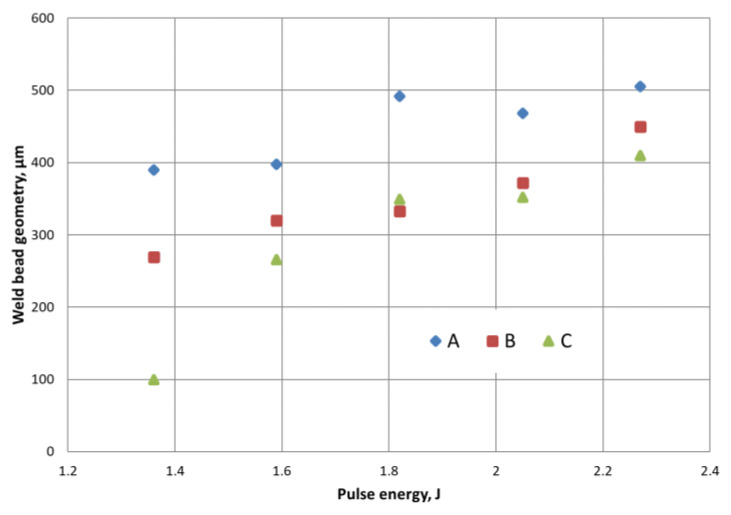
Dependence of characteristic dimensions of the penetration on the pulse energy value. Geometrical parameters A, B and C according to Figure 2.

**Figure 6 materials-14-03213-f006:**
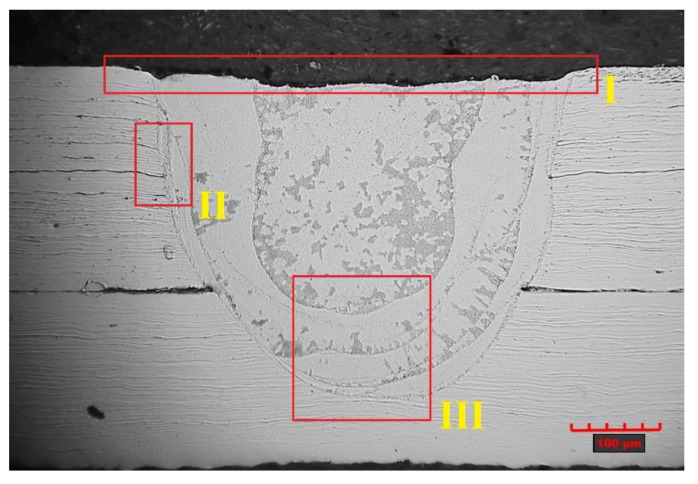
Structure of the weld with the characteristic areas of the joint marked. I – weld face, II and III - the fusion penetration line of the weld into the base material.

**Figure 7 materials-14-03213-f007:**
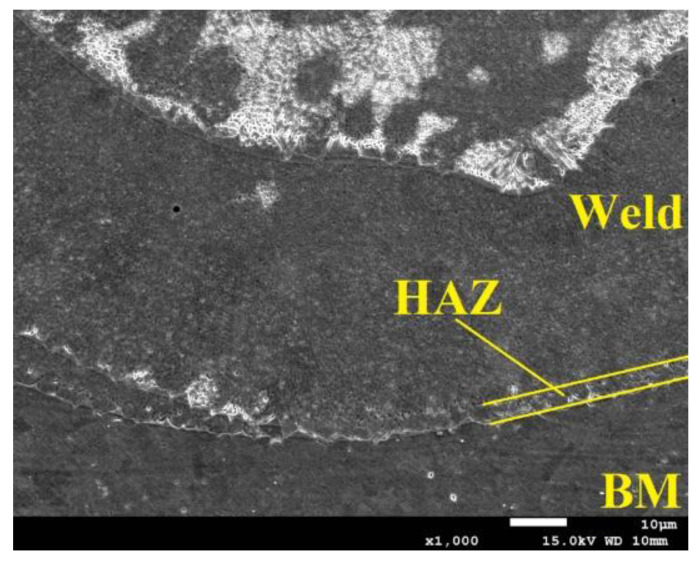
The fusion penetration line zone of the weld into the base material.

**Figure 8 materials-14-03213-f008:**
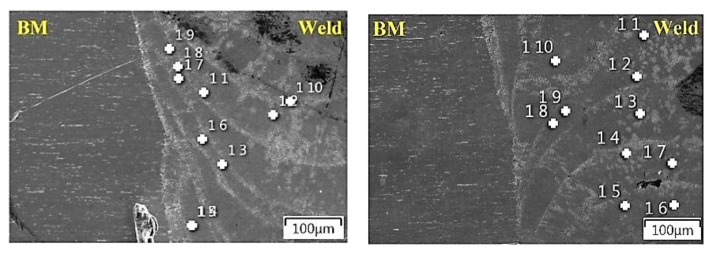
Measurement points of the spectrum analysis and the result of the readings for point 11, for a weld made without a protective atmosphere (**A**), and for point 12, for a weld made in an argon shroud (**B**).

**Figure 9 materials-14-03213-f009:**
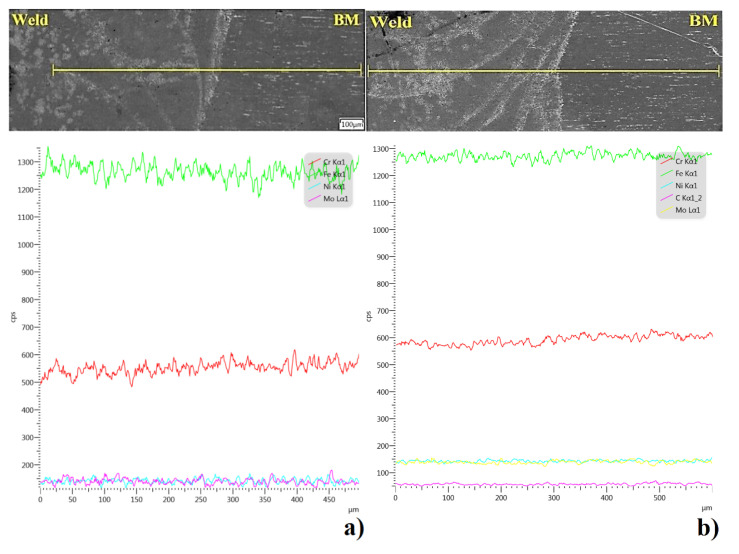
Linear analysis of the distribution of elements (Cr, Fe, Mo, Ni) in the weld–MR transition zone: (**a**) a weld made in air; (**b**) a weld made in an argon shroud.

**Figure 10 materials-14-03213-f010:**
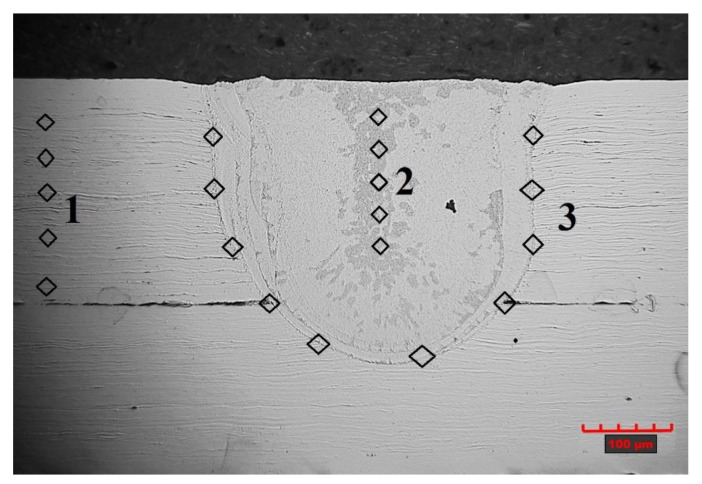
The scheme of microhardness measurements: the tube core material (1); the central part of the penetration (2); near the fusion penetration line (3).

**Figure 11 materials-14-03213-f011:**
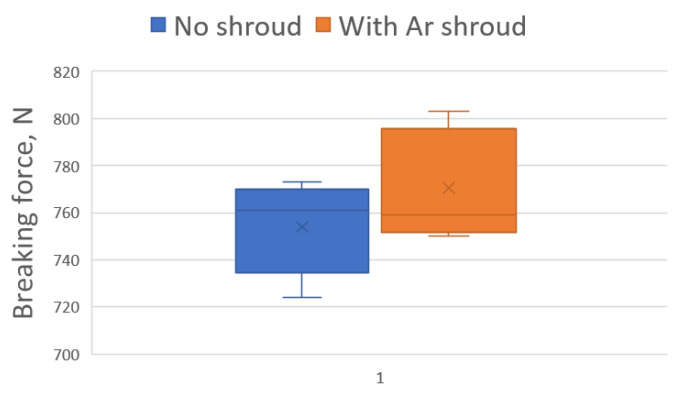
Results of tensile shear test.

**Figure 12 materials-14-03213-f012:**
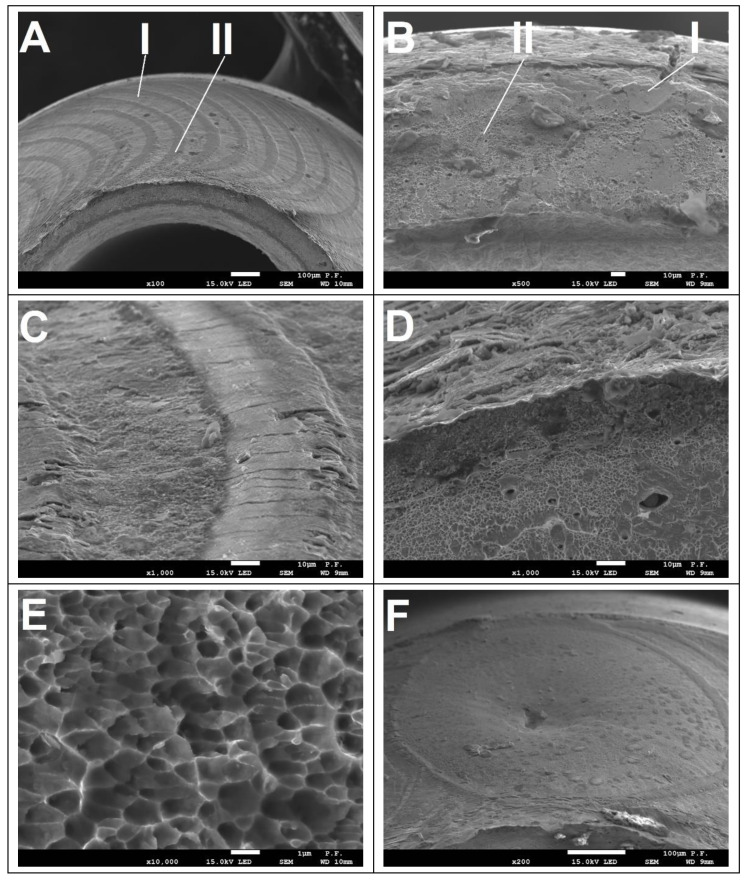
Morphology of the sample fracture after breaking: (**A**) 500× magnification; (**B**) 100× magnification; (**C**) 1000× magnification; (**D**) 500× magnification; (**E**) 10,000× magnification; (**F**) 200× magnification.

**Table 1 materials-14-03213-t001:** Chemical composition of 316 L steel according to EN10204 certified 190821-S047.

%C	%Si	%Mn	%P	%S	%Cr	%Ni	%Mo	%N
0.009	0.485	1.615	0.0307	0.0041	16.763	11.208	2.041	0.0619

**Table 2 materials-14-03213-t002:** List of welding parameters accepted after visual testing.

Weld No.	Power Setting %	Pulse Energy (J)	Pulse Power (kW)	Welding Linear Energy (J/mm])
1	7	1.36	0.34	8.69
2	8	1.59	0.4	10.13
3	9	1.82	0.45	11.58
4	10	2.05	0.51	13.03
5	11	2.27	0.57	14.47

**Table 3 materials-14-03213-t003:** Percentage content of selected alloying elements at measurement points.

Welding Atmosphere	Measurement Point	Si %	Mn %	Cr %	Ni %	Mo %
in air	1 1	0.31	1.06	16.75	10.26	2.54
	1 2	0.32	1.07	16.76	10.49	2.42
in argon	11	0.39	1.25	16.78	10.40	2.45
	12	0.26	0.86	16.9	10.58	2.15

**Table 4 materials-14-03213-t004:** Tensile shear test results of the fabricated micro-joints.

Sample No	Sisma No Shroud (N)	Sisma With Ar Shroud (N)
Sample 1	773	759
Sample 2	767	788
Sample 3	724	753
Sample 4	761	803
Sample 5	745	750
Mean value (N)	754	770.6
Standard deviation (N)	19.74	23.56

## Data Availability

Not applicable.

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
