# Peer review of "Technology and Properties of Peripheral Laser-Welded Micro-Joints"

_materials, 2021, doi:10.3390/ma14123213_

Round 1
Reviewer 1 Report
The reviewed article deals with micro-joints of thin walled AISI 316L pipes manufactured by laser welding. The list of detailed remarks is given below:
- Line 54-55 - please join the sentence.
- Fig. 1 - is it own Authors photo? Was it published? There is a lack of a), b) and c) on the figures.
- Table 2 - please change the unit of the welding linear energy.
- Fig. 5 - please make the legend more visible.
- Please add the scheme of microhardness measurements.
- Table 4 - is it a tensile strength or rather tensile shear one from a point of view of the joint? For the results, please make a standard accuracy.
- Fig. 11 - please add the legend.
- Fig. 12 - which sample is presented?
- There is no discussion of the results - it is the main weakness of the article.
- Please add to the discussion following paper: DOI: 10.3390/ma13132930.
Reviewer 2 Report
In this manuscript the authors present the results of research on the technology and peripheral properties of laser-welded micro-joints. An important goal of this study was to determine the characteristics of properly made joints of thin-walled AISI 316L steel tubes as well as to indicate the range of optimal parameters of the welding process by using Nd: YAG laser. The joints shall have a weld with no visible welding imperfections, therefore, the selection of parameters treatment is important.
The presented manuscript of the authors are interesting. The some issue are not clearly described in manuscript. Therefore before publishing should be consider the following comments:
- Table 2 should contain the exact chemical composition of the steel.
- The title of the table should be changed to “Chemical composition of 316L steel”
- It would be worth to mark what is visible in Figure 3, because the Figure 3 is composed of two pictures. Please markt the basic elements of the laser device structure in the pictures.
- Figure 4 is not to scale. Please complete this.
- The scale in Figure 6 is unreadable.
- It would be better to compare the spectra from Figures 8 and 9 if they had a similar scale.
- Graph for the strength of welds made without argon shroud on Figure 11 have not a legend.
Reviewer 3 Report
Introduction
General Remark: A lot of investigations were performed in the past considering laser based micro welding. Additionally, the metallurgy of AISI 316L is completely understood. This authors should clearly point out the novelty of their research. What is the research hypothesis? In this context, the state of the art is completely missing in the introduction. This must be improved.
Figure 1: The scale is not at all visible
Experimental Tests
Line 90ff: The testing must be detailed, e.g. microhardness (load,...), EDS analysis (keV, ....), location of cross-sections, etc.....
Table 2: The pulse duration is missing
Line 149ff: In the eyes of the reviewer, the description "pre-tests" should part of the chapter "Results".
Figure 4: The quality of the picture must be improved. The red marked regions should be detailed by means of pictures with a higher magnification
Figure 5: In Figure 5, the following should be added "Geometrical parameter A, B and C according to Figure 2"
Line 184ff: Based on the results......the structure of welds 1 and 5......: This description should be underlined by a higher magnification of the welds 1 and 5
Figure 10: The picture must be improved. The quality is unacceptable
Line 284-292: This paragraph should be part of the "experimental testing"
Line 293ff: The results are shown in Table....: A picture of the fractured samples is necessary to point out the location of the fracture (weld, base metal)
Figure 11: Force should be replaced by Tension [N/mm^2] and elongation [%]
Line 316ff: This brittle fracture behaviour is untypical for austenites. It seems to me that there was an incomplete fusion between the weld metal and base material. According to Figure 12 B - Zone I, no penetration or fusion is existent.
Reviewer 4 Report
This study dealt with the technology developed for the laser assisted micro-welding for circumferential joining hollow tube. I would like to have the authors address the following issue in the revised submission:
- In Abstract: Please highlight the key outcome with quantifiable results.
- In line 42-44: The authors mentioned the dissimilar material joints which is out of the context of this study. Please provide relevant literature review.
- Figure 1: Scale is not readable. Please modify images accordingly.
- Figure 4: Please provide scale in Figure 4.
- Figure 10: Scale in Y- axis in (a) and (b) should be same to avoid confusion. Please modify the figures.
- Please avoid redundancy in mentioning the instrument model/name beginning of each subsection in Section 3. These have been already mentioned in experimental section.
- In Summary (Line 367-368): The authors stated, “avoiding the formation of surface oxides when welding in argon shroud”. However, there was no evidence of oxidation reported in this study while welding was performed without argon shroud. This is the major weakness of this manuscript. I would like the authors to revisit their microstructural analyses section (Line 179-283) to address the issue.
- The authors should perhaps take assistance from the technical editing service to improve the readability in English overall. Few places including but not limited to: Lines 52-53, 127-128,
Round 2
Reviewer 3 Report
unfortunately, most of my remarks were not considered....According to the answers of the reviewers, I noticed the following
1 The scale on Figure 1 was changed but is still not at all visible for the reader
2 Experimental Tests: According to the authors, the introduction of the experimental testing was changed. The reviewer is not able to retrace the performec changes. Important information such as microhardness (load,..), EDS analysis (keV,..) is still missing...
3 OK
4 The authors left this record unchanged. Why? Please, change.
5 OK
6 OK
7 OK
8 OK
9 The authors left this record unchanged...Why? Please, change according to my remark.
10 OK
11 The authors mentioned that original Figure 11 has been changed. However, this Figure is not existent anymore and replaced by a new one. The y-axe was defined as Strength in N. This is wrong. Either Strength in N/mm² or Force in N.
12 OK
